# Predicting Impacts of Climate Change on Northward Range Expansion of Invasive Weeds in South Korea

**DOI:** 10.3390/plants10081604

**Published:** 2021-08-05

**Authors:** Sun Hee Hong, Yong Ho Lee, Gaeun Lee, Do-Hun Lee, Pradeep Adhikari

**Affiliations:** 1School of Plant Science and Landscape Architecture, Hankyong National University, Anseong-si 17579, Gyeonggi-do, Korea; shhong@hknu.ac.kr; 2Institute of Ecological Phytochemistry, Hankyong National University, Anseong-si 17579, Gyeonggi-do, Korea; yongho@korea.ac.kr (Y.H.L.); skyge723@gmail.com (G.L.); 3OJeong Resilience Institute, Korea University, Seongbuk-gu, Seoul 02841, Korea; 4National Institute of Ecology, Seocheon-gun 33657, Chungcheongnam-do, Korea; eco0407@nie.re.kr

**Keywords:** climate change, invasive weeds, MaxEnt, South Korea, suitable habitat

## Abstract

Predicting the distribution of invasive weeds under climate change is important for the early identification of areas that are susceptible to invasion and for the adoption of the best preventive measures. Here, we predicted the habitat suitability of 16 invasive weeds in response to climate change and land cover changes in South Korea using a maximum entropy modeling approach. Based on the predictions of the model, climate change is likely to increase habitat suitability. Currently, the area of moderately suitable and highly suitable habitats is estimated to be 8877.46 km^2^, and 990.29 km^2^, respectively, and these areas are expected to increase up to 496.52% by 2050 and 1439.65% by 2070 under the representative concentration pathways 4.5 scenario across the country. Although habitat suitability was estimated to be highest in the southern regions (<36° latitude), the central and northern regions are also predicted to have substantial increases in suitable habitat areas. Our study revealed that climate change would exacerbate the threat of northward weed invasions by shifting the climatic barriers of invasive weeds from the southern region. Thus, it is essential to initiate control and management strategies in the southern region to prevent further invasions into new areas.

## 1. Introduction

Invasive species are non-indigenous species that are introduced outside their native ranges either deliberately or inadvertently and can threaten native biodiversity, alter the structure and function of ecosystems, disrupt natural and agricultural landscapes, and result in large-scale economic losses [1,2,3,4]. Invasive species alter the dynamics of plant communities by reducing the amounts of nutrients, water, and space available to native species and by changing the soil chemistry, hydrological pattern, and moisture-holding capacity in the region of invasion [5,6,7,8,9].

Climate change exacerbates threats to, and losses of, biodiversity through multiple mechanisms, including reductions in climatic barriers and the facilitation of the spread of invasive species [5,6]. Therefore, there is great concern regarding the introduction, establishment, and naturalization of invasive species as a result of climate change and the consequent problems caused by invasive species in natural ecosystems worldwide [10,11]. Over the last century, the global temperature has increased by 0.78 °C, and it is predicted to increase by 2.6 °C to 4.8 °C by 2100 [12]. The rate of climate change in South Korea is projected to be higher than that of the rest of the world [13]. In South Korea, the average temperature has increased 1.8 °C over the last 100 years, and it is predicted to increase 1.75 °C temperature by 2050, 2.35 °C by 2070, and 5.7 °C by 2100 compared to the 19881~2005 average under the representative concentration pathway (RCP) 8.5 scenario [13]. Thus, various environmental problems, including the spread of invasive species, are expected to be particularly serious in South Korea.

Predicting the potential distributions of invasive species under current and future climate change conditions will help prioritize the species and locations to target for early detection and will help leaders adopt the best preventive measures [14]. Hence, there is a pressing need to develop reliable methods for predicting invasions ahead of time so that surveillance or management policies can be adjusted to reduce the risk of the establishment, spread, and overall expansion of invasive species [15]. Species distribution models (SDMs) are considered the most reliable technique for simulating the range of suitable habitats of native and invasive species in response to climatic and environmental variables [9,16]. More recently, SDMs have been widely used in the fields of ecology, conservation biology, biogeography, and natural resource management [11,16,17,18,19,20]. Among the various available SDMs, the maximum entropy (MaxEnt) model is a frequently used machine learning technique that can obtain high predictive performance with a limited species presence and environmental variable data [21,22,23,24]. The MaxEnt model is well known for modeling invasive species because it only uses species presence data; absence data for invasive species may not always be reliable because the ranges of the species may be expanding and not yet have reached equilibrium, which may lead to incorrect interpretation [25].

Among the various types of invasive species, the types of plants that are considered invasive weeds include small seasonal herbs to intermediate perennial shrubs growing on open and degraded lands, transportation corridors, riversides, and seashores [11,26,27,28]. Anthropogenic activities, such as the development, maintenance, and expansion of roads and railway connections, trade and tourism, and natural processes that occur via air, water, and wild animals, are known to be major vectors for the dispersal of invasive weeds across the world, including in and to South Korea [6,11,29]. In South Korea, crop fields, orchards, pastures, and forests are at a high risk of invasion by invasive weeds [27,30,31,32,33,34]. Altogether, 320 taxa of invasive and alien plants were listed in South Korea in 2016, of which more than 95% are invasive weeds. This includes the 16 ecologically most disruptive weeds, such as *Rumex acetosella*, *Paspalum distichum*, and *Conyza canadensis* [28,30,31]. Invasive alien species have been reported to cause economic damage of 22.6 billion Korean Won (KRW; approximately 19.6 million USD as of July 2021) and the government of South Korea invests approximately 5 billion KRW (approximately 4.3 million USD as of July 2021) per year in the control and management of invasive species [35].

To maintain the control and management of invasive weeds, regular studies are required to understand the ecology and distribution of these invasions, as well as their impacts on agriculture and forestry. Although many ecological studies have been carried out to investigate invasive species [27,28,31,32,33,36], very few studies have addressed plant invasions under climate change in South Korea [24,37,38]. The southern region of South Korea, including Jeju Province, Jeollanam Province, Gyeongsangnam Province, Busan City, and Ulsan City, is recognized as a high-risk region for the introduction and establishment of tropical and subtropical invasive weeds [30,31]. It is essential to identify areas that are potentially at risk of invasion under future climate change in the central and northern regions of the country. Therefore, this study was designed to assess the habitat suitability of the 16 most disruptive weeds [31] in agricultural and natural ecosystems that are mainly distributed in the southern region of South Korea, including Jeju Island, under projected bioclimatic scenarios (RCP 4.5 and RCP 8.5) and land cover changes. Although 320 invasive weeds have been recorded in South Korea, we were able to collect the minimum number of species occurrence records required for MaxEnt modeling [39] for the 16 most problematic weeds, and consequently, only these species are used in this study. Species richness was estimated in different regions and across the country. This study will provide fundamental information about current and future potential habitats, which could be useful for developing control and management strategies for invasive weeds.

## 2. Results

### 2.1. Variable Selection and the Importance of Variables in Model Performance

The Pearson’s correlation test was performed to select the important and independent variables among the 19 bioclimatic variables and two environmental variables analyzed in this study for modeling purposes (Appendix A). Based on the weak correlations (r < 0.60) with each other, six bioclimatic variables and two environmental variables were selected: the annual mean temperature (Bio01), isothermality (Bio03), temperature seasonality (Bio04), annual precipitation (Bio12), precipitation of the wettest month (Bio13), precipitation of the driest month (Bio 14), distance from water (d-water), distance from roads (d-road), and land cover (SSP1) (Table 1). Therefore, these nine variables were selected for use in MaxEnt modeling of invasive weeds.

The contributions of these bioclimatic and environmental variables to the model performance varied among the studied invasive weeds (Table 2). Bio04 had the highest contribution to ten species (35.46–79.61%), including *Astragalus sinicus*, *Gnaphalium calviceps*, and *Chenopodium ambrosioides*; the Bio01 had the highest contribution to four species (59.72–82.73%), including *Bromus unioloides* and *Coronopus didymus*; and Bio12 had the highest contribution to two species one of which was *Sisyrinchium angustifolium* and another was and *Spergularia rubra* (78.56%). These results indicate that temperature seasonality, annual mean temperature, and annual precipitation played major roles in affecting the distribution of invasive weeds, while other variables only played minor roles. Similarly, Jackknife test was performed (Appendix A) for checking the relative contribution of environmental variables for each invasive weed, which showed contribution of variables were varied among the weeds.

The model performance was evaluated using the AUC and TSS values (Table 3). The average AUC value of the 16 studied invasive weeds was 0.982 ± 0.018, ranging from 0.943 (*Lolium multiflorum*) to 0.998 (*Silene gallica*), and the mean TSS value was 0.908 ± 0.083, ranging from 0.780 (*A. sinicus*) to 0.998 (*Spergularia rubra*). The ROC curve for each invasive weed is shown in Appendix A. Similarly, the average value of Kappa was 0.64 ± 0.065, ranging from 0.547 (*A. sinicus*) to 0.735 (*C. didymus*). These AUC values show that the model performs reliably in predicting the habitat suitability of invasive weeds, and that the outputs of the model are very close to the approximation of the true probability distribution. Similarly, the TSS and Kappa values show that the observations and predictions of the model are in agreement and also show the accuracy of the model when predicting suitable habitats using only presence data. 

### 2.2. Prediction of Habitat Suitability under Current and Future Climate Change Scenarios

The extent of habitat suitability for the 16 studied invasive weeds was modeled to show the distribution of each species (Appendix A) and the estimated suitable habitat areas under current and future climate change scenarios (Table 4). The predicted suitable areas for ten invasive weeds, such as *Modiola caroliniana* (658 km^2^), *G*. *calviceps* (942 km^2^), and *Paspalum dilatatum* (1503 km^2^), covered less than 3% of the land area of South Korea under the current climate conditions. However, the suitable areas for three weeds, including, *B*. *unioloides* (17,319 km^2^), *S. officinale* (10,706 km^2^) and *A. sinicus* (16,375 km^2^), covered more than 10% of the land area nationwide. The model shows that suitable habitat areas for all invasive weeds will increase in the future. The rate of increase in suitable habitat area was not uniform for all species. The rate of increasing habitat suitability was estimated to be 49.23–6115.46% by 2050, and 145.19–12,611.36% by 2070 under RCP 4.5, compared to current suitable habitat.

A habitat suitability map of invasive weeds under current climatic conditions is presented in Figure 1. Moderately suitable and highly suitable habitat areas for invasive weeds are presently concentrated in the southern region of South Korea, for example, in Jeju Province, Jeollanam Province, and Gyeongsangnam Province. However, the central region (for example, Chungcheongnam Province, Daejeon Province, and Gyeongsangbuk Province) and northern regions (for example, Gyeonggi Province, Gangwon Province, and Seoul City) exhibit extremely large areas with marginally suitable habitats. The areas of marginally suitable, moderately suitable, and highly suitable habitats for invasive weeds across the country were estimated to be 85,112.02 km^2^, 8877.46 km^2^, and 990.29 km^2^, respectively, under the current climate conditions (Table 5).

Future climate change will increase both the extent and intensity of habitat suitability for invasive weeds in South Korea. 

The predicted moderately suitable habitat and highly suitable habitat areas were highest under RCP 8.5, with the exception of moderately suitable habitat for 2070, and these habitat categories were predicted to exist in all provinces of the central region by 2050 (except Chungcheongbuk Province and Sejong Province), and in all provinces of the northern region by 2070. These results showed that RCP 8.5 would be more favorable for the habitat expansion of invasive weeds than RCP 4.5. We added areas of moderately suitable and highly suitable invasive weed habitats and estimated the percentages of suitable areas in terms of the total surface area of each region. Under RCP 4.5, the proportion of suitable areas will be 96.97 to 99.18 in the southern region, 62.42 to 75.08 in the central region, and 9.00 to 36.13 in the northern region between the current period and 2070 (Figure 2). Similarly, under RCP 8.5, the proportion of suitable areas will range from 99.08 to 99.9 in the southern region, 85.61 to 88.33 in the central region, and 27.81 to 60.31 in the northern region between the current period and 2070 (Figure 2). These results showed that climate change is likely to facilitate the expansion of suitable invasive weed habitats.

### 2.3. Prediction of Species Richness under Current and Future Climates

The potential species richness values of invasive weeds under current and future climate change scenarios are presented in Figure 3.

Under current climatic conditions, the average and maximum species richness values across the country were estimated to be 1.65 and 13, respectively. Under climate change, the average richness values were predicted to increase up to 369.48% by 2050 and 605.15% by 2070. However, the maximum species richness was not estimated to increase as much as the average species richness. The maximum species richness values were predicted to increase up to 15.38% by 2050 and up to 23.07% by 2070. 

To understand the trend of increasing species richness from the southern region to the northern region, we estimated the average and maximum richness values of the studied invasive weeds, as can be seen in Table 6. This shows that the current species richness was most pronounced in the southern region and was lower in the central and northern regions. In the future, the rate of increase in species richness will reach a maximum in the northern region (55,457%), followed by those in the central region (4284%) and southern region (247%) by 2070. However, the maximum species richness will be similar in the southern region and will increase up to 87.5% in the central region and up to 250% in the northern region by 2070.

### 2.4. Cluster Analysis of Invasive Weeds

The PCA determined the ordering of the samples and showed three distinctive groups (groups 01, 02, and 03) of invasive weeds with similar distributions and spread potential in the PCA plot (Figure 4). The invasive weeds clustered in groups 01, 02, and 03 had small, intermediate, and large distribution areas, respectively. Group 01 consisted of three species, including *A*. *leptophyllum*, *S*. *gallica* and *C*. *didymus*; group 02 consists of five species, including *G*. *calviceps*, *S. angustifolium*, *S. rubra*, *M. caroliniana* and *S*. *rhombifolia*, and group 03 consists of four species, including *A*. *sinicus* and *B*. *unioloides*, under both RCP 4.5 and RCP 8.5. However, four species, including *O. laciniate*, *P*. *dilatatum*, *Malva parviflora* and *C*. *ambrosioides*, clustered in different groups under either RCP 4.5 or RCP 8.5. We performed the Kruskal–Wallis test to examine the significant differences among the groups using the Dwass–Steel–Critchlow–Fligner method. This showed that there was a significant difference (*p* < 0.001) among groups 01, 02, and 03 across the country in 2050 and 2070 (Table 7). However, there was no significant difference among the groups in the southern, central, and northern regions at the current time, or in the northern region in 2050 (Table 7). 

## 3. Discussion

Our study investigated the potential habitats of 16 problematic invasive weeds that are currently found in the southern region of South Korea using the MaxEnt modeling approach. The current suitable habitats for the studied invasive weeds that were predicted by the model were highly matched with the current existing records. The average AUC (0.982 ± 0.004), TSS (0.881 ± 0.881), and Kappa (0.64 ± 0.065) values indicated that the model performance was excellent, and that perfect agreement existed between the observations and predictions [41,42]. In this study, temperature seasonality, annual mean temperature, and annual precipitation were the dominant driving factors for the determination of the habitats of the studied invasive weeds. Temperature seasonality balances photosynthesis and regulates the growth, reproduction, and other physiological functions of plants [43]. 

Similarly, increasing the annual mean temperature and annual precipitation may create suitable habitats for invasive weeds while altering the distribution and abundance of existing native species, which reduces competition with local species [44]. Therefore, these variables could be critical in determining the spread of invasive weeds in the future. Similar cases were reported by Adhikari et al. [24] and Wang et al. [45] for predicting the habitat expansion of invasive and alien plants. However, the other variables examined in this study only had minor contributions to the model. 

Climate change may directly or indirectly influence the introduction, dispersion, and establishment of invasive and alien species, and may decrease the resilience of natural ecosystems to invasive species [6]. Climate change may also modify the geographical ranges and environmental impacts of invasive species, as well as the economic costs necessary for their management [46]. Our model shows that climate change is likely to substantially increase the habitat suitability of invasive weeds in the southern region of South Korea. These results reinforce the spatially explicit evidence that supports the earlier hypothesis that warming temperatures will expand the suitable habitats of invasive plants northwards [47]. This is also evidenced by previous observations and projections on the impacts of warmer climatic conditions [5,24,48,49]. For most of the studied invasive weeds, habitat suitability will expand toward the central and northern regions of South Korea under climate change, and this impact will be particularly visible in Chungcheongnam Province, Chungcheongbuk Province, Gyeongsangbuk Province, Gyeonggi Province, and Seoul City, which are up to 454 km from the southern region (for example, Jeju Province and Jeollanam Province), by 2070. Consistent with our findings, several studies have attempted to model the habitat suitability of non-native and invasive species in South Korea and across the globe and have projected the expansion of their ranges northwards in response to climate change [5,9,23,24,50]. Some studies on invasive plants showed that the majority of range shifts that were expected to occur by 2070 will occur as early as 2050 in Europe [51]. To our knowledge, this is the first study to describe the potential impacts of climate change and land cover change on these invasive weeds in South Korea. This study forms part of our ongoing project. We will compare the results obtained in this study with those obtained from other ecological niche models.

In this study, we predicted the habitat suitability of 16 problematic invasive weed species. The rate and extent of habitat suitability were not projected to be consistent among all invasive weed species. *A*. *sinicus*, *B*. *unioloides*, *C*. *ambrosioides* and *S*. *officinale* var. *leiocarpum* are estimated to have relatively high habitat suitability areas, covering 78.44% and 86.91% of the land area of the country by 2050, and 2070 (RCP 4.5), respectively; these values are comparable to those reported in previous studies performed in South Korea [24,33,37]. The PCA revealed that the studied invasive weeds could be divided into three groups based on similarities in their invasion potentials and distribution patterns. The invasive weeds included in groups 01, 02, and 03 had low, intermediate, and high invasion potentials and distribution patterns, respectively. Group 01 is currently found in Jeju Province (Jeju Island) and is projected to have very limited habitat expansion along the coastal side of the southern region, for example in Gyeongsang Province, indicating that continental climatic features such as large diurnal and seasonal temperature ranges, low annual precipitation, and low relative humidity may not favor the expansion of these species [52]. However, the invasive weeds present in groups 02 and 03 are projected to expand continuously toward the central and northern regions with various rates of invasion, indicating that these species could have greater tolerance to continental climates.

In comparison to other regions, the average and maximum species richness values calculated in the southern region, especially in Jeju Province, were estimated to be the highest under current climatic conditions. Currently, the average species richness in the southern region is estimated to be 11.30, and this region has been invaded by a maximum of 15 invasive weeds, including *Sida rhombifolia*, *S*. *gallica* and *Spergularia rubra*; however, the northern region, for example, Gangwan Province, has estimated average and maximum richness values of 0.014 and 4, respectively. These results showed that the introduction, establishment, and dispersion of the studied invasive weeds occur more readily in the southern region. The climatic conditions of this region are characterized by a warm-temperate and humid climate, which favors invasive weeds originating from the tropical and subtropical climates of South America, southern Europe, China, and South Asia [31]. Usually, invasive weeds that are indigenous to tropical and subtropical countries have much higher critical thermal maxima than native species, suggesting that these invasive species can thrive at higher temperatures and may successfully outcompete native species under climate change [6]. Under current climatic conditions, the average winter temperature in the southern region is approximately 3 °C, which could favor the survival of warm-adapted invasive weeds in the winter season. However, in the central and northern regions, the average winter temperatures reach approximately −10 °C, which could limit the distribution of such invasive weeds, as described by Hou et al. [53] and Petitpierre et al. [54]. With increasing temperatures under climate change, suitable habitats for invasive weeds will expand toward the central and northern regions of South Korea due to the removal of current climatic barriers and the shifting of plant hardiness zones northward [6,55], and consequently, species richness is predicted to increase in the future. In addition, climate change will negatively affect native species and ecosystems by changing their phenology, composition, distribution, and adaptability through changes in environmental conditions and by creating difficulties that prevent native species from surviving and competing with invaders, which could be conducive to invasive species taking over newly empty niches [6,56].

These future changes in habitat suitability depend not only on the climatic temperature and precipitation variables used in the model, but also on many non-climatic factors, including land topography, soil and habitat characteristics, and on the morphological and physiological advantages of individual plant species, such as short life cycles, high fecundity, strong dispersal abilities, and phenotypic plasticity, which allow them to survive adverse climate conditions [5,23,57,58]. Thus, under the same climatic conditions in the future, the climatically suitable habitat for each invasive weed species could be different. Many invasive weeds are characteristically recognized as broadly ecologically and environmentally tolerant, such as *S*. *gallica* and *S*. *officinale* are salt-resistant and can grow in coastal areas and in very disturbed areas along roadsides [59,60]. Similarly, some invasive weeds, such as *Apium leptophyllum*, *M. caroliniana*, *C*. *ambrosioides* and *S**ida rhombifolia*, grow in diverse habitats, including croplands, farmlands, riversides, and dry and drained lands [27,28,31,32]. Therefore, the suitable habitats for all invasive weeds will expand in the future.

The studied invasive weed species were all introduced to South Korea either intentionally or unintentionally (Table 8). *A*. *sinicus* is native to China and was intentionally introduced to South Korea before 1937 to improve pastures; the species subsequently invaded grasslands and interior forests [31]. Similarly, *B*. *unioloides* and *Lolium multiflorum* were introduced from South America and southern Europe for use as cattle-feed, and have since become invaders of grasslands, forests, and crop lands [28,31]. Other invasive weeds were unintentionally introduced in South Korea from different countries in America, Europe, and Asia, probably via foreign trade, tourism, and tidal activity in the seas [24,28,30,31], and many anthropogenic activities, such as road construction, land cover changes, and the importation of agricultural seeds from foreign countries have accelerated their invasion rates. Invasive weeds have adverse impacts on agricultural and wild ecosystems through increased labor input for weeding, reduced crop production, the replacement of the native forage of cattle and wild herbivores such as roe deer (*Capreolus pygargus*) [61], and their negative effects on forest ecosystems [62]. Therefore, the economic losses and negative impacts of invasive weeds on food security, biodiversity, and ecosystem services in the near future could increase considerably if control and preventive measures are not adopted in time.

Although this study provided valuable information about the potential habitat suitability of invasive weeds in different provinces of South Korea under current and future climatic conditions, our models were dependent on bioclimatic and some environmental variables, such as land cover change, distance from the road, and distance from water and is disregarding, other important predictors such as land topography, soil characteristics, dispersal capacities, biotic interaction (e.g., facilitation and competition), and vectors driving species invasions, as described by Pysek and Richardson [63] and Buri et al. [64]. This study is a part of ongoing research; we would consider using other variables, including topographic and soil characteristics, dispersion capacities, and biotic interactions, to obtain more accurate predictions in the near future.

## 4. Materials and Methods

### 4.1. Study Area and Species Data

This study was carried out on the mainland and all islands of South Korea, which constitutes the southern portion of the Korean Peninsula, with a total land mass of 100,033 km^2^ (Figure 5). The geography of South Korea is largely mountainous and comprises 70% of the Korean Peninsula, with mountains in the north and east and lowlands and flat plains in the south and west. The climate of South Korea is categorized into warm-temperate, temperate, and cold-temperate climate types in the southern, central, and northern regions and high mountains, respectively [65].

The southern part of the country is hot and humid, while the northern part is cold and continental. The average winter temperature for the study area ranges from −6 °C to 3 °C, and the average summer temperature ranges from 23 °C to 26 °C [66]. The annual precipitation ranges from 1000 to 1800 mm and varies greatly from summer to winter, with a higher rate of precipitation in the southern regions, including Jeju Island, than in the northern and central regions [66]. The vegetation is broadly classified into temperate broadleaf, deciduous broadleaf, coniferous, subalpine, and alpine [67]. Altogether, 41,483 species have been reported to constitute the overall biodiversity, including 5308 plant species, 22,612 invertebrate species, and 1899 vertebrate species [67].

We divided the study area into three regions, based on the latitude: the southern region (<36° N latitude), which includes seven provinces (Jeju, Jeollanam, Gwangju City, Gyeongsangnam, Jeollabuk, Busan City, and Ulsan City); the central region (approximately between 36° N and 37° N latitude), which includes six provinces (Ghungcheonnam, Sejong, Daejeon, Chungcheongbuk, Daegu City, and Gyeongsangbuk); and the northern region (>37° N latitude), which includes four provinces (Gyeonggi, Seoul City, Incheon City, and Gangwon).

The 16 most problematic invasive weeds [31] which mainly occur in the southern region of South Korea, were selected to study rapid range expansion (Table 8). Species occurrence points were recorded through field surveys performed between March 2014 and November 2020, and additional occurrence points were collected from various published reports [27,28,30,32]. Since invasive weeds were most prevalent adjacent to roads, we recorded species occurrence adjacent to roads using a Garmin GPS unit (GPSmap 64sx). The species occurrence survey, plot design, and survey method were performed according to the guidelines of the National Institute of Ecology, South Korea (NIE). The multiple species occurrence points in the same grid of ~1 km^2^ spatial resolution were removed and retained a single unique point per grid by using spatially rarefy occurrence data tool in Arc GIS SDM tool box 2.4 [68] to minimize the overfitting and incorrect inflation of the model outputs due to spatial autocorrelation [69]. After spatial filtering, the total species occurrence points comprising the 16 most problematic invasive weeds were reduced from 2304 to 1487, and these were used in modeling. The species occurrence points for each invasive weed are shown in Figure 6. 

### 4.2. Bioclimatic and Environmental Variables

We considered bioclimatic variables [70,71], distance from water, distance from roads, and land cover to be important parameters that affect the distribution of invasive weeds in South Korea. Therefore, we collected climatic data, including precipitation and monthly minimum and maximum temperatures, from the Korea Meteorological Administration (KMA) to estimate the current and future climate change scenarios in South Korea. We selected two greenhouse gas emission scenarios, widely known as RCP 4.5 and RCP 8.5, for 2050 and 2070. RCP 4.5 and RCP 8.5 represent the moderate and highest emission scenarios, corresponding to projected global mean surface temperature increases of 1.4 °C to 1.8 °C and of 2.0 °C to 3.7 °C, respectively [12]. Global circulation models (GCMs) consider the physical processes of the atmosphere, earth surface, ocean, and cryosphere [72]. They capture the underlying processes that respond to climate forcing, for example, concentrations of greenhouse gases, surface albedo changes, aerosols, and solar irradiance [50,72]. Globally, GCMs are constantly being updated by different modeling groups to incorporate higher spatial resolution, biogeochemical cycles, and new physical processes. One such GCM is the HadGEM3-RA regional atmospheric model that was developed by the Met Office Hadley Centre (https://www.metoffice.gov.uk/, accessed on 3 August 2021) based on the atmospheric component of the latest Earth System Model [73,74]. The KMA has utilized HadGEM3-RA [73] and prepared a national climate change scenario for South Korea. HadGem3-RA has a tendency to model small-scale features more realistically than other GCMs, for example, HADGEM2-AO, owing to its high resolution, which includes complicated topography, long and irregular coastlines, and thousands of islands in the Korean Peninsula. Therefore, we used HadGEM3-RA GCM to develop the climate change scenarios RCP 4.5 and RCP 8.5 using the Dismo package in GNU R [75]. This GCM has been used in various studies for modeling indigenous and invasive plant species in South Korea [19,24,49,76].

The current climate was estimated by averaging the climatic data recorded from between 1950 and 2000, and the future climatic conditions in 2050 and 2070 were estimated from the projections for 2046 to 2055 and for 2066–2075, respectively. Each climatic dataset had a spatial resolution of 0.01° (30″) or approximately 1 km^2^. In addition to bioclimatic variables, we used maps that provided data for MaxEnt modeling that included seven land cover categories (agricultural land, grassland, urban land, forest land, barren land, wetland, and water), distance from roads, and distance from water. We used the current and future land cover change scenarios (shared socioeconomic pathway, SSP 1) obtained from the Korea Environment Institute, MOTIVE (www.kaccc.kei.re.kr, accessed on 3 August 2021), as developed by Song et al. [40]. South Korea has thousands of rivers, streams, lakes, reservoirs, and ponds. Many invasive weeds, such as *A*. *leptophyllum*, grow in disturbed areas along these water resources. Rivers and streams play important roles in the seed dispersal of PIWs. Similarly, roads function as prime habitats and corridors for the spread of invasive weeds. The design and maintenance of roads, and resultant increase in human activities, can accelerate invasion and displace native species [11,77]. Once invasive weeds have become established along roadsides, they could function as a source of invasions into adjacent cropland, grassland, and forest [29]. Therefore, we developed two variables, distance from water (d-water) and distance from roads (d-road), using the Euclidian distance function of ArcGIS 10.3 (Esri, Redlands, CA, USA) with the same resolution of 1 km^2^ as was used for the other bioclimatic variables. We performed Spearman’s correlation on pairs using the Proc Corr function of SAS 9.4 (SAS Institute Inc., Cary, NC, USA) to eliminate the autocorrelation (r^2^ > 0.75, *p* = 0.05) among the bioclimatic and environmental variables and selected the nine variables with high predictive performance, as in Shin et al. [19] and Adhikari et al. [18] (Appendix A).

### 4.3. Model Development

The current and future suitable habitats for invasive weeds in South Korea were predicted using the machine learning algorithm MaxEnt package version 1.3.3 for GNU R (https://cran.r-project.org/src/contrib/Archive/maxent/, accessed on 3 August 2021). MaxEnt is a widely used habitat suitability modeling technique. It exhibits a high predictive performance using only presence data [16] and is most commonly used for invasive species [21,24,78]. MaxEnt is considered a good option for invasive species because presence-absence survey data are rarely available, as is the case in this study. Additionally, MaxEnt can provide reliable estimates of potentially suitable habitats for invasive species at small spatial scales, even with limited data [79,80].

As the MaxEnt model required background data (e.g., pseudo-absence), we used 15,050 background points selected randomly throughout the entire study area using ArcGIS 10.3, as suggested by Barbet-Massin et al. [81]. The models were calibrated using 75% of the species occurrence points (presence and background points) and validated by the remaining 25% [82]. The other options for the model were run with default settings, and the model was replicated ten times.

### 4.4. Model Evaluation and Validation

The goodness of fit of the model was evaluated, and the area under the curve (AUC) values of the receiver operating characteristic (ROC) curves [83], the true skill statistic (TSS) [41], and Kappa values [84] were used for model validation. The AUC values, TSS values, and Kappa values were computed using the test data points. To test the model results, the AUC was used as a threshold-independent technique to differentiate presence from absence; the AUC value ranges between 0 and 1 and is used to evaluate the performance of a model [85]. The AUC value is independent of the size of the dataset (prevalence). However, this metric is sometimes criticized because it equally weights commission and omission errors, and this can provide false predictions [86], especially when the study area is small. It gives a high AUC (overfitting), which can mislead the model evaluation. A higher AUC value suggests a superior performance of the model. In this study, the model performance as assessed based on the AUC value was graded as failing (0.5–0.6), average (0.6–0.7), good (0.7–0.8), very good (0.8–0.9), or outstanding (0.9–1) [42]. The TSS (sensitivity + specificity–1) accounts for both commission and omission errors but is unaffected by the prevalence. Consequently, this metric has been used as an alternative criterion for the validation of model efficiency [41,87]. The TSS evaluates the outputs of a model by examining the classification accuracy after a threshold value has been selected. Similarly, the Kappa statistic measures prediction accuracy in comparison to what could have been achieved by chance alone [41]. The TSS and Kappa values ranged between −1 and +1, where +1 indicates perfect agreement between the observations and predictions, and a value of 0 or less indicates a performance no better than random [41,88]. Therefore, we used the AUC, TSS, and Kappa values to assess the model performance. The database and detailed methodological flowchart of this study are shown in Figure 7.

### 4.5. Image Combination, Classification and Analysis

The habitat suitability maps of 16 problematic invasive weeds under current and future climate change scenarios (RCP 4.5 and RCP 8.5) for 2050 and 2070 were summed using the Raster 3.4 package in GNU R 4.03. Based on the logistic scale, the current and future habitat suitability maps of invasive weeds were classified into three categories: marginally suitable, suitable, and highly suitable habitats. We selected pixels with a value greater than 0.5, to consider areas that represent at least 50% probability of species occurrence. Thus, the value greater than 0.5 (>50%) represent areas with highly suitable habitat, while values between 0.1 to 0.5 (10–50%) represent moderately suitable, and value lower than 0.1 (<10%) represent marginally suitable habitat for invasive weeds, which are very similar to the division of habitat suitability by Thapa et al. [89]. The area of each category was estimated using the raster calculator of the spatial analyst tool in ArcGIS 10.3. The areas of moderately suitable and highly suitable habitats were summed, and the percentage change in the suitable habitat areas in the southern region (<36° N latitude), central region (36–37° N latitude), and northern region (>37° N latitude) of South Korea were determined. Similarly, the average and maximum species richness values of the studied invasive weeds were estimated at the national level and at three regional levels using the zonal statistics function of the spatial analyst tool in ArcGIS 10.3.

### 4.6. Cluster Analysis of Invasive Weeds

We performed principal component analysis (PCA) using the potential distribution area of invasive weeds in different regions of South Korea under the current climate, and under the future climate change scenarios (RCP 4.5, RCP 8.5) for 2050 and 2070, to describe differences in distributions and ordering of the samples, as per Dyderski et al. [90]. Then, we used the Kruskal–Wallis test [91] with pairwise two-sided multiple comparison analysis via the Dwass–Steel–Critchlow–Fligner method (*p* > 0.05) to test the significance of the difference among the means of the groups obtained by the PCA.

## 5. Conclusions

In this study, we estimated the habitat suitability of invasive weeds under current and future climate conditions in South Korea. The suitable habitat areas of all invasive weeds were predicted to expand in the future, while retaining their current ecological niches. Currently, the southern region of South Korea has the highest proportion of climatically suitable areas for all invasive weeds, and these areas are predicted to expand toward the central and northern regions under climate change. Invasive weeds have already had negative impacts on biodiversity, food security, livelihoods, and ecosystem services in South Korea. Thus, the effective management of invasive weeds is required using integrated approaches that combine mechanical, chemical, and biological controls at the early stage of invasive weed species population establishment. Therefore, these findings are important in informing the development of control strategies for invasive weeds at local and regional scales. Based on our study, we believe that early detection and eradication strategies, including mowing and soil seed removal strategies, are needed in the southern region, especially in Jeju Province, Jeollanam Province, Gyeongsangnam Province, Busan City, and Ulsan City of South Korea.

## Figures and Tables

**Figure 1 plants-10-01604-f001:**
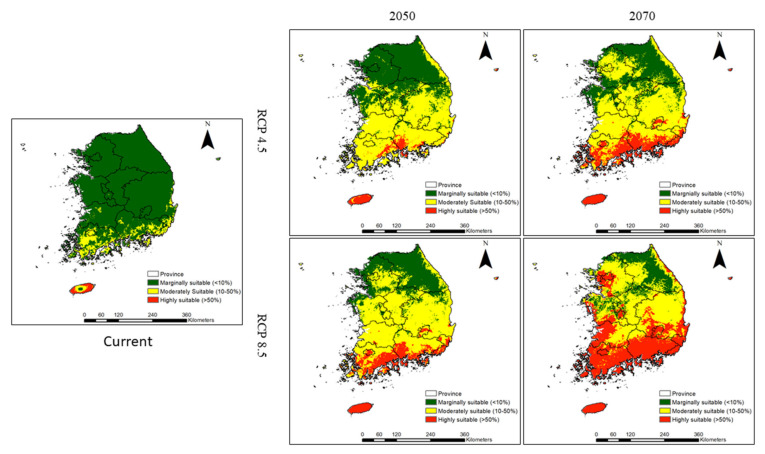
Prediction of the habitat suitability of invasive weeds under current climate conditions and under future climate change scenarios RCP 4.5 and RCP 8.5.

**Figure 2 plants-10-01604-f002:**
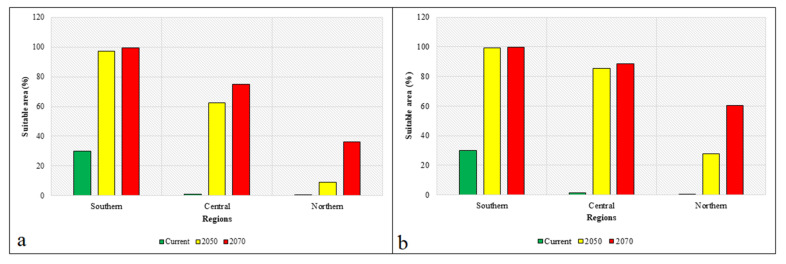
Changes in the suitable areas for invasive weeds in different regions of South Korea under the current climate conditions and under the future climate change scenarios RCP 4.5 (**a**) and RCP 8.5 (**b**) by the years 2050 and 2070.

**Figure 3 plants-10-01604-f003:**
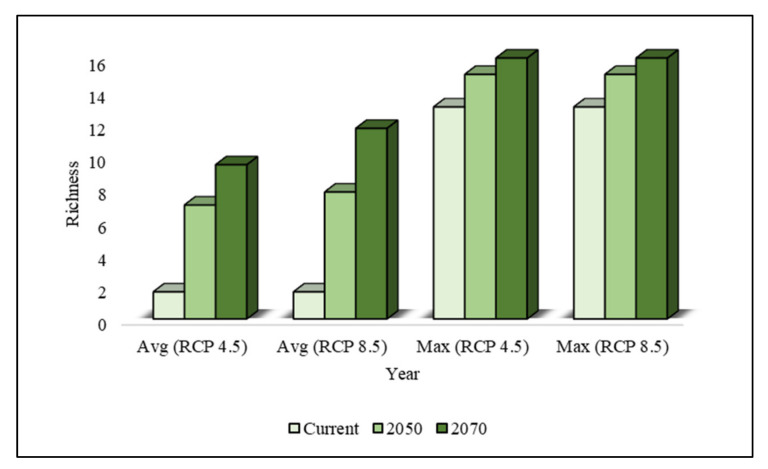
Average and maximum species richness of invasive weeds in South Korea under the current and future climate change scenarios (RCP 4.5 and RCP 8.5).

**Figure 4 plants-10-01604-f004:**
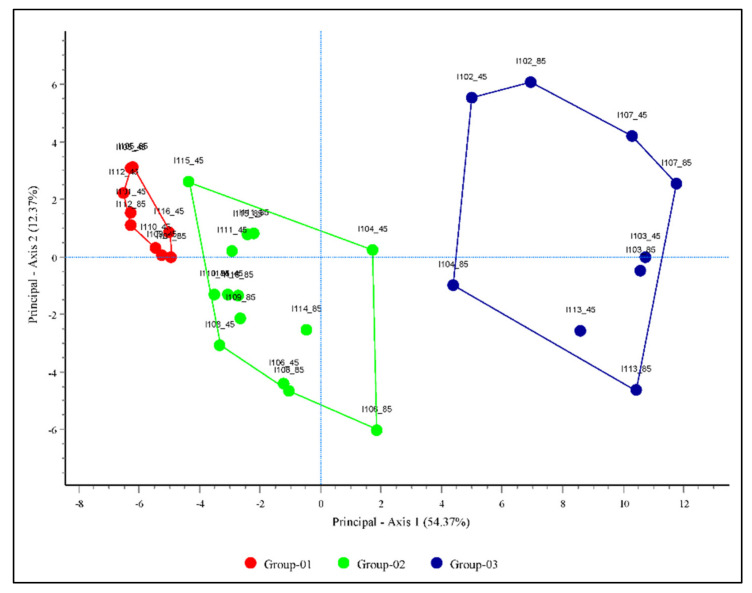
Principle component analysis using potential distribution area of invasive weeds under the current climate and future climate change scenarios RCP 4.5 and RCP 8.5 for the years 2050 and 2070. Groups 01, 02, and 03 represent invasive weeds with low, medium, and high invasion potentials, respectively. Detailed information on the invasive weeds corresponding to those shown in the PCA plot is provided in Table 8.

**Figure 5 plants-10-01604-f005:**
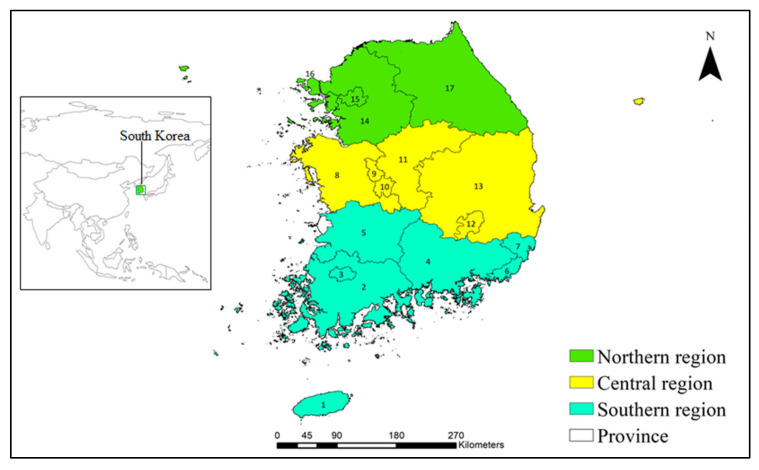
Map of the study area. The three different colors indicate the southern, central, and northern regions of South Korea, as shown in the legend. The numbers 1–16 indicate the different provinces of South Korea. 1, Jeju; 2, Jeollanam; 3, Gwangju City; 4, Gyeongsangnam; 5, Jeollabuk; 6, Busan City; 7, Ulsan City; 8, Ghungcheonnam; 9, Sejong City; 10, Daejeon City; 11, Chungcheongbuk; 12, Daegu City; 13, Gyeongsangbuk; 14, Gyeonggi; 15, Seoul City; 16, Incheon City; and 17, Gangwon.

**Figure 6 plants-10-01604-f006:**
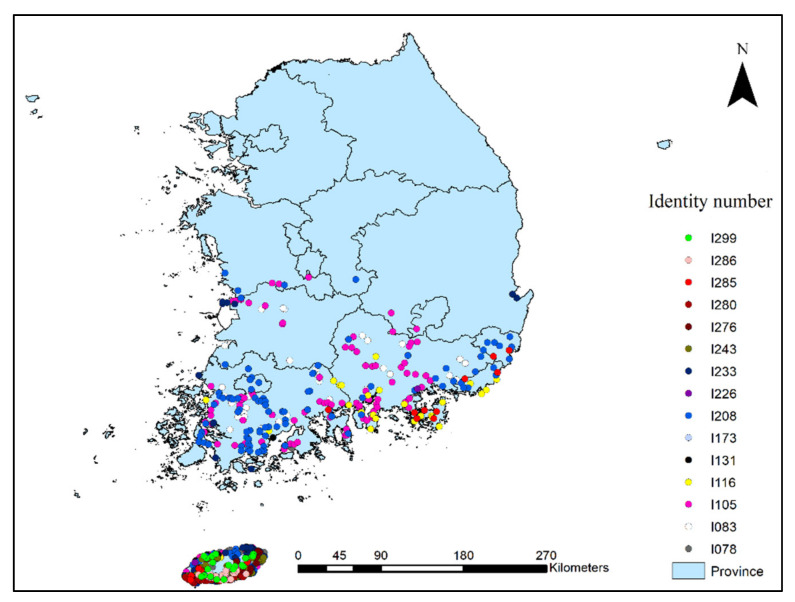
Species presence points for the studied invasive weed species. The legend shows the identity number of each invasive weed species. I101, *Apium leptophyllum*; I102, *Astragalus sinicus*; I103, *Bromus unioloides*; I104, *Chenopodium ambrosioides*; I105, *Coronopus didymus*; I106, *Gnaphalium calviceps*; I107, *Lolium multiflorum*; I108, *Modiola caroliniana*; I109, *Oenothera laciniate*; I110, *Paspalum dilatatum*; I111, *Sida rhombifolia*; I112, *Silene gallica*; I113, *Sisymbrium officinale*; I114, *Sisyrinchium angustifolium*; I115, *Spergularia rubra*; I116, *Malva parviflora*. Details of these invasive weeds are given in Table 8.

**Figure 7 plants-10-01604-f007:**
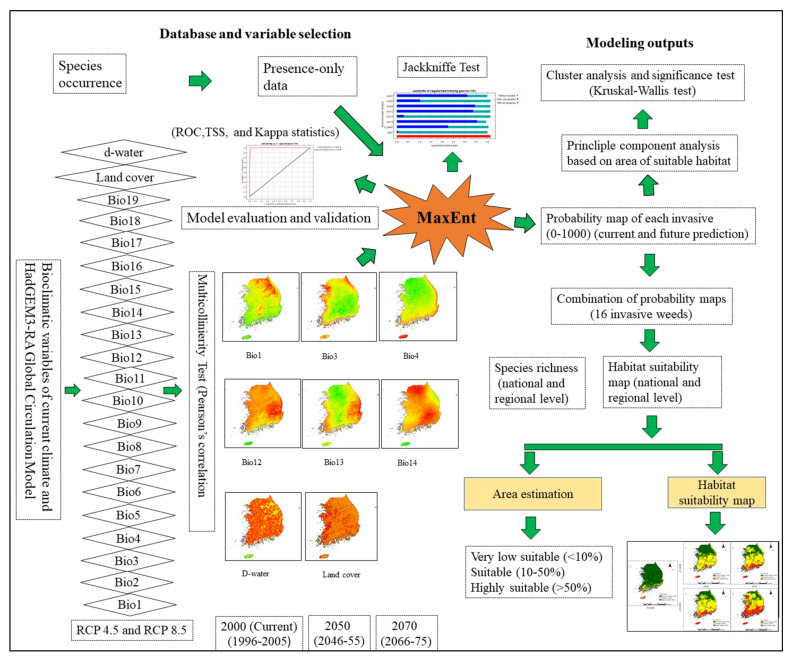
Flow chart of development and building blocks of MaxEnt modeling, and its practical application for the estimation and prediction of habitat suitability of invasive weeds that are currently present in southern region of South Korea.

**Table 1 plants-10-01604-t001:** List of variables used in the modeling of invasive weeds.

Code	Description	Unit	Source
Bio01	Annual mean temperature	Degrees Celsius	KMA
Bio03	Isothermality (BIO2/BIO7) (*100)	Percentage	KMA
Bio04	Temperature seasonality	Percentage	KMA
Bio12	Annual precipitation	Millimeters	KMA
Bio13	Precipitation of wettest month	Millimeters	KMA
Bio14	Precipitation of driest month	Millimeters	KMA
d-water	Distance from water	Meters	This study
d-road	Distance from roads	Meters	This study
SSP1	Land cover	-	Song et al. [40]

KMA = Korea Meteorological Administration. *, Multiplication sign

**Table 2 plants-10-01604-t002:** Contribution of bioclimatic and environmental variables to models.

Name of Species	Bio1	Bio3	Bio4	Bio12	Bio13	Bio14	d-Roads	d-Water	Land Cover
*Apium leptophyllum*	18.27	0.00	54.00	0.87	0.00	22.85	0.39	1.61	2.00
*Astragalus sinicus*	26.71	6.89	35.46	11.84	0.06	3.05	0.12	2.25	13.61
*Bromus unioloides*	65.21	2.41	12.06	10.13	0.24	2.95	0.42	0.68	5.91
*Chenopodium ambrosioides*	7.58	0.60	67.51	20.42	0.00	0.74	0.28	0.50	2.37
*Coronopus didymus*	63.70	0.00	4.19	7.64	0.04	5.00	0.01	14.36	5.05
*Gnaphalium calviceps*	1.90	0.01	57.69	6.71	0.03	1.27	0.16	31.54	0.68
*Lolium multiflorum*	59.72	1.56	19.22	8.66	0.49	4.29	0.74	0.90	4.43
*Modiola caroliniana*	13.47	1.10	64.17	6.60	0.98	0.00	0.00	11.20	2.48
*Oenothera laciniata*	12.71	0.26	45.98	28.11	0.00	11.37	0.02	0.94	0.62
*Paspalum dilatatum*	0.43	0.01	59.58	26.99	0.00	12.02	0.37	0.17	0.42
*Sida rhombifolia*	82.73	0.54	0.12	0.97	0.33	4.70	1.62	8.39	0.60
*Silene gallica*	2.31	1.51	79.61	12.20	0.00	0.00	2.86	1.51	0.00
*Sisymbrium officinale*	32.19	0.79	62.79	0.20	0.00	0.00	3.87	0.10	0.04
*Sisyrinchium angustifolium*	6.38	1.10	43.43	46.65	0.00	0.52	1.63	0.02	0.28
*Spergularia rubra*	2.79	0.00	2.55	78.56	0.00	14.53	0.38	0.39	0.79
*Malva parviflora*	6.37	0.00	72.98	4.07	0.83	0.00	5.07	8.05	2.64

**Table 3 plants-10-01604-t003:** AUC, TSS, and Kappa values for each invasive weed.

Species Name	Number of Occurrences	AUC Value	TSS Value	Kappa Value
*Apium leptophyllum*	78	0.996	0.922	0.666
*Astragalus sinicus*	70	0.956	0.78	0.547
*Bromus unioloides*	303	0.948	0.748	0.603
*Chenopodium ambrosioides*	93	0.977	0.842	0.619
*Coronopus didymus*	94	0.997	0.987	0.735
*Gnaphalium calviceps*	81	0.996	0.974	0.669
*Lolium multiflorum*	173	0.943	0.79	0.651
*Modiola caroliniana*	72	0.996	0.951	0.646
*Oenothera laciniata*	115	0.994	0.958	0.635
*Paspalum dilatatum*	69	0.993	0.974	0.691
*Sida rhombifolia*	58	0.994	0.958	0.56
*Silene gallica*	38	0.998	0.985	0.570
*Sisymbrium officinale*	35	0.978	0.861	0.521
*Sisyrinchium angustifolium*	116	0.992	0.959	0.723
*Spergularia rubra*	44	0.994	0.998	0.700
*Malva parviflora*	48	0.97	0.853	0.713

**Table 4 plants-10-01604-t004:** Area of suitable habitats in km^2^ for the studied invasive weeds in South Korea.

Name of Species	Current	RCP 4.5		RCP 8.5	
		2050	2070	2050	2070
*Apium leptophyllum*	1467	2191	3597	3597	20,677
*Astragalus sinicus*	16,375	59,873	59,873	46,413	63,288
*Bromus unioloides*	17,319	70,045	69,498	67,783	87,762
*Chenopodium ambrosoides*	6819	44,756	57,690	56,419	67,152
*Coronopus didymus*	1000	9698	9862	8783	9417
*Gnaphalium calviceps*	942	38,213	77,125	61,615	83,264
*Lolium multiflorum*	10,441	54,688	49,746	50,182	88,962
*Modiola caroliniana*	658	15,865	35,216	12,899	62,166
*Oenothera laciniata*	2060	16,585	22,113	15,153	43,731
*Paspalum dilatatum*	1503	10,585	11,394	7503	29,986
*Sida rhombifolia*	1984	53,154	72,819	65,412	81,971
*Silene gallica*	1667	4616	5090	6466	10,905
*Sisymbrium officinale*	10,706	74,504	82,553	78,303	93,290
*Sisyrinchium angustifolium*	2674	35,800	64,329	35,379	77,379
*Spergularia rubra*	194	12,058	24,660	30,308	49,970
*Malva parviflora*	3192	14,750	19,474	18,255	44,151

**Table 5 plants-10-01604-t005:** Predicted areas of marginally suitable, moderately suitable, and highly suitable invasive weed habitats in km^2^.

Scenario	Year	Marginally Suitable Area	Moderately Suitable Area	Highly Suitable Area
	Current	85,112.02	8877.46	990.28
RCP 4.5	2050	39,486.1	51,105.08	4388.597
	2070	26,776.6	52,956.23	15,246.96
RCP 8.5	2050	25,322.82	57,256.83	12,400.12
	2070	15,140.92	42,811.42	37,027.43

**Table 6 plants-10-01604-t006:** Average and maximum species richness values of invasive weeds in different regions.

Scenario	Year	Southern Region		Central Region		Northern Region	
	Richness	Average	Maximum	Average	Maximum	Average	Maximum
	Current	3.83	15.00	0.21	8.00	0.01	4.00
RCP 4.5	2050	10.48	15.00	4.81	14.00	2.99	15.00
	2070	11.25	15.00	6.90	15.00	4.71	14.00
RCP 8.5	2050	10.79	15.00	5.35	15.00	3.66	13.00
	2070	13.29	16.00	9.36	15.00	8.14	16.00

**Table 7 plants-10-01604-t007:** Average distribution and significance test among three groups invasive weeds using Kruskal–Wallis Test.

Country/Region	Year	Average Area (km^2^ ± S.E.)			
		Group 01	Group 02	Group 03	K–W Test
Total	Current	1039 ± 198 C	2163 ± 1126 BC	15,322 ± 2928 AB	*
	2050	4875 ± 880 C	28,489 ± 6271 B	11,2538 ± 17,906 A	***
	2070	11,206 ± 2307 C	53,721 ± 7356 B	131,753 ± 20,314 A	***
Southern	Current	1031 ± 200 C	1934 ± 901 BC	13,934 ± 2837 A	*
	2050	4537 ± 871 C	22,487 ± 4253 B	54,106 ± 6340 A	***
	2070	10,058 ± 1961 C	33,162 ± 3870 B	55,026 ± 6056 A	***
Central	Current	6 ± 5 C	212 ± 212 BC	1371 ± 328 AB	*
	2050	159 ± 78 C	1940 ± 627 B	47,115 ± 8862 A	***
	2070	963 ± 325 C	12,008 ± 1638 B	52,880 ± 8560 A	***
Northern	Current	2 ± 1	17 ± 16	18 ± 15	N.S.
	2050	179 ± 61 C	4062 ± 2284 BC	11,317 ± 3492 AB	*
	2070	185 ± 59 C	8551 ± 2894 B	23,847 ± 6568 AB	***

*, *p* < 0.05; ***, *p* < 0.001; N.S., not significant; S.E., standard error; K–W test, Kruskal–Wallis test; the letter A, B, and C used to show difference among the groups.

**Table 8 plants-10-01604-t008:** List of invasive weeds used in species distribution models.

Id No.	Scientific Name	Common Name	Growth Form	Native Range	Mode of Introduction	Introduction Period	Degree of Naturalization
I101	*Apium leptophyllum*	Marsh parsley	WA	Australia	Unintentional	1964–2010	IV
I102	*Astragalus sinicus*	Chinese milkvetch	BA	China	Intentional	Before 1937	II
I103	*Bromus unioloides*	Rescue grass	P	South America	Intentional	Before 1999	IV
I104	*Chenopodium ambrosioides*	Mexican tea	SA	South America	Unintentional	Before 2000	II
I105	*Coronopus didymus*	Swine wartcress	WA	Europe	Unintentional	Before 1996	II
I106	*Gnaphalium calviceps*	Narrowleaf purple everlasting	BA	South America	Unintentional	1964–2010	IV
I107	*Lolium multiflorum*	Italian ryegrass	WA	Europe and Africa	Intentional	Before 2000	III
I108	*Modiola caroliniana*	Red flowered mallow	BA	South America	Unintentional	Before 1980	I
I109	*Oenothera laciniata*	Cutleaf evening primrose	BA	North America	Unintentional	Before 1980	IV
I110	*Paspalum dilatatum*	Dallas grass	P	South America	Unintentional	1964–2010	IV
I111	*Sida rhombifolia*	Queensland-hemp	P	South Asia	Unintentional	Before 1980	II
I112	*Silene gallica*	Common catchfly	BA	Europe	Unintentional	Before 1996	IV
I113	*Sisymbrium officinale*	Hedge mustard	WA	Europe	Unintentional	Before 1999	I
I114	*Sisyrinchium angustifolium*	Blue-eyed grass	P	USA	Unintentional	Before 1999	
I115	*Spergularia rubra*	Red sandspurry	BA	Eurasia	Unintentional	Before 1996	I
I116	*Malva parviflora*	Cheese weed	SA	North Africa	Unintentional	Before 2000	I

BA, biannual; P, perennial; SA, summer annual; WA, winter annual I–IV indicates degree of naturalization: I, rare or uncommon; II, distributed over a small area and at low density; III, moderately distributed and at medium density; IV, widely distributed and at high density.

## Data Availability

Not applicable.

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
