# Peer review of "Predicting Impacts of Climate Change on Northward Range Expansion of Invasive Weeds in South Korea"

_plants, 2021, doi:10.3390/plants10081604_

Round 1

Reviewer 1 Report

Comments on Plants manuscript #1298863;

Predicting impacts of climate change on northward range expansion of invasive weeds in South Korea; S-H Hong, Y H Lee, G Lee, D-H Lee, P Adhikari

This manuscript describes a study of the potential effects of climate change on the future ranges of 16 invasive weed species in South Korea. The authors use the modeling software Maximum Entropy to establish the environmental factors associated with the current locations of the species, then use that relationship to predict where those conditions will occur with a changing climate. This software and methodology have been used for study of climate change for many other species. The manuscript is well written and except for one issue, my comments mostly address editorial questions. I suggest the authors should provide additional explanation for the variable “distance from water” and how it is an important factor affecting the presence of suitable habitat for a weed species (see my comment below for line 197).

L23 - Define the acronym "RCP" in abstract

L48 - Provide reference for the statement "The rate of climate change in South Korea ....."

L120 - Not clear what geographic area the stated winter and summer temperatures apply to -  perhaps the study area. If that is correct, consider changing the sentence to read: "For the study area, the average winter temperature ... "

L140 - Text "of invasive" used twice.

L143 - Please define the acronym NIE.

L197 - Please provide a more complete description of the variable "distance from water." For example, is water defined in ArcGIS 10.3 as a lake, river, or ocean? Also, is there a minimum size for the water variable, such as 1km2? Is a small mountain stream considered to be the same as a large river in a valley? How is this variable an important factor related to the distribution of a weed species?

L650 - The citation for reference 25 shows Adhikari, P. listed twice in the author list; is that correct?

L720 - Incomplete citation for reference NIMS.

L724 - This citation for reference 51 shows Adhikari, P. listed twice; is that correct?

L743 - Verify spelling "andfFuture"

Reviewer 2 Report

Intellectually the manuscript plants-1298863 is very complete. The authors analyzed potential scenarios of the risk of weed expansion in South Korea under climate change conditions, clearly describing which species will have the best situational conditions to invade new areas.

Intellectually I found no weaknesses, but throughout the manuscript there are many typos. In the attached document the authors can find my suggestions, mainly to the use of decimals and avoid writing in the second person and use of subjective terms. The reference list has many typographical errors. So the authors must correct carefully.

Reviewer 3 Report

Dear authors,

I have made several comments and suggestions, which I hope you will find useful and constructive.

Round 2

Reviewer 3 Report

Dear authors,

Thank you for addressing all the issues raised. I have no further comments to make.